# CUL4-Based Ubiquitin Ligases in Chromatin Regulation: An Evolutionary Perspective

**DOI:** 10.3390/cells14020063

**Published:** 2025-01-07

**Authors:** Makiko Nakagawa, Tadashi Nakagawa

**Affiliations:** 1Institute of Gene Research, Yamaguchi University Science Research Center, Yamaguchi 755-8505, Japan; mnakagaw@yamaguchi-u.ac.jp; 2Advanced Technology Institute, Life Science Division, Yamaguchi University, Yamaguchi 755-8611, Japan; 3Division of Cell Proliferation, United Centers for Advanced Research and Translational Medicine, Graduate School of Medicine, Tohoku University, Sendai 980-8575, Japan; 4Department of Clinical Pharmacology, Faculty of Pharmaceutical Sciences, Sanyo-Onoda City University, Sanyo-Onoda 756-0084, Japan

**Keywords:** CUL4, CRL4, ubiquitin, ubiquitin ligase, chromatin, evolution

## Abstract

Ubiquitylation is a post-translational modification that modulates protein function and stability. It is orchestrated by the concerted action of three types of enzymes, with substrate specificity governed by ubiquitin ligases (E3s), which may exist as single proteins or as part of multi-protein complexes. Although Cullin (CUL) proteins lack intrinsic enzymatic activity, they participate in the formation of active ubiquitin ligase complexes, known as Cullin-Ring ubiquitin Ligases (CRLs), through their association with ROC1 or ROC2, along with substrate adaptor and receptor proteins. Mammalian genomes encode several CUL proteins (CUL1–9), each contributing to distinct CRLs. Among these CUL proteins, CUL1, CUL3, and CUL4 are believed to be the most ancient and evolutionarily conserved from yeast to mammals, with CUL4 uniquely duplicated in vertebrates. Genetic evidence strongly implicates CUL4-based ubiquitin ligases (CRL4s) in chromatin regulation across various species and suggests that, in vertebrates, CRL4s have also acquired a cytosolic role, which is facilitated by a cytosol-localizing paralog of CUL4. Substrates identified through biochemical studies have elucidated the molecular mechanisms by which CRL4s regulate chromatin and cytosolic processes. The substantial body of knowledge on CUL4 biology amassed over the past two decades provides a unique opportunity to explore the functional evolution of CRL4. In this review, we synthesize the available structural, genetic, and biochemical data on CRL4 from various model organisms and discuss the conserved and novel functions of CRL4s.

## 1. Introduction

The active maintenance of protein homeostasis through coordinated protein synthesis and degradation is vital for organismal health [1,2,3]. In eukaryotic cells, protein degradation occurs via the lysosome or proteasome. As the lysosome resides in the cytosol, nuclear protein degradation is predominantly facilitated by the proteasome. The proteasome typically identifies polyubiquitylated proteins as substrates, making precise protein polyubiquitylation crucial for proper protein degradation, particularly in the nucleus.

Ubiquitylation functions not only as a degradation signal but also as a regulator of protein functionality. The versatility of protein ubiquitylation arises, in part, from the diversity of chain types formed by ubiquitin (mono-, multi-, and polyubiquitylation) and the various linkage types (M1, K6, K11, K27, K29, K33, K48, and K63) [4,5,6,7]. Moreover, proteomic analyses have identified approximately 90,000 ubiquitylation sites, surpassing the ~15,000 acetylation sites and ~20,000 phosphorylation sites found in mammalian cells [8,9], underscoring the significance of this modification.

Ubiquitylation is orchestrated by the sequential activity of three classes of enzymes: ubiquitin-activating (E1), conjugating (E2), and ligating (E3) enzymes. Ubiquitin ligases (E3s) confer substrate specificity in ubiquitylation by either mediating the transfer of ubiquitin from the E2 ubiquitin-conjugating enzyme to the substrate or directly accepting ubiquitin on their own cysteine residue from the E2 enzyme before transferring it to the substrate. Some ubiquitin ligases function as single proteins, while others form multi-protein complexes comprising E2-binding proteins and substrate-recognizing proteins. The notable examples of multi-protein ubiquitin ligases include Cullin-Ring ubiquitin Ligases (CRLs), where Cullin (CUL) proteins bind to ROC1 or ROC2, E2 enzyme linker proteins, and substrate adaptor and receptor proteins, thereby constituting active ubiquitin ligases [10] (Figure 1a). Through a diverse array of substrate receptors, CRLs are estimated to catalyze as much as 50% of cellular protein ubiquitylation in Trypanosoma brucei [11] and approximately 20% of ubiquitylation in cultured mammalian cells [12].

Evolutionarily, it is proposed that the eukaryotic ancestor harbored three Cullin genes (*CULα*, *CULβ*, and *Culγ*, each analogous to *CUL1*, *CUL3*, and *CUL4* in mammals, respectively), forming the foundation of all extant Cullins, though lineage-specific expansions have been observed in some species (Figure 1b). For example, yeast genomes encode three Cullin genes (*CUL1*, *CUL3*, and *CUL4*), whereas mammalian cells express eight CUL proteins (CUL1, 2, 3, 4A, 4B, 5, 7, and PARC), with CUL2, CUL5, CUL7, and PARC emerging from the CUL1 prototype [13,14,15], and CUL4 undergoing duplication to generate CUL4A and CUL4B [16,17,18].

Compared with CUL4A, CUL4B contains N-terminal extension where nuclear localization signal is located. Therefore, CUL4B resides in the nucleus, whereas CUL4A is primarily observed in the cytosol [19,20]. The nuclear localization of CUL4A is promoted under DNA-damaged conditions [21,22], thereby facilitating CUL4A’s engagement in nuclear functions. Given that the amino-terminal 100 amino acids, which serve as the binding site for DDB1, are critical for the nuclear localization of CUL4A [22], it is hypothesized that DDB1 and DCAF proteins play a contributory role. Nonetheless, the precise molecular mechanism underlying this process remains unidentified. Notably, the single CUL4 proteins in Trypanosoma, plants, yeasts, worms, and flies all show nuclear localization [15,23,24,25,26,27,28]. Consistently, genetic and molecular evidence strongly implicates an ancient role for CUL4 in chromatin regulation, while also suggesting that CUL4 acquired a cytosolic function in vertebrates with the emergence of a cytosol-localizing paralog, CUL4A. The identification of nearly 100 substrates has clarified the detailed molecular mechanisms underlying CRL4 function. In this review, we summarize our current understanding of CUL4, derived from structural, genetic, and molecular studies, with an emphasis on its role in chromatin function.

## 2. CRL4 Structure

CRL4 is composed of three invariant proteins—the CUL4 scaffold protein, the ROC1 E2 enzyme linker protein, and the DDB1 substrate adaptor protein—along with variable substrate receptors, collectively termed DCAF (DDB1 and CUL4-associated factors) or DWD (DDB1-binding WD40 domain) proteins, due to the presence of a WD40 domain in most cases [29] (Figure 1a). The ubiquitin ligase activity of CRL4 is regulated by the covalent attachment of NEDD8 (Neddylation) on a single lysine residue of CUL4 [30]. Thus, NEDD8-harboring CRL4 is able to associate with ROC1 and DCAFs, but its removal is required for ROC1 and DCAF releases, which are required for DCAF recycling. COP9 signalosome (CSN) is responsible for NEDD8 removal (deneddylation), and its inactivation leads to the dysfunction of CRL4 [31] (Figure 2).

The crystal structures of CRL4A with a viral protein SV5V [32], CRL4A/B^DDB2^ [33], CRL4A^DCAF1^ [34], and CRL4A^CSB^ [35,36] have been resolved. These studies collectively reveal that CUL4 forms an elongated and rigid structure, with the ROC1-binding site located at the C-terminal region and the DDB1-binding site at the N-terminal region (Figure 2). DDB1 is divided into three β-propeller domains (BPA, BPB, and BPC), with BPB interacting with an α-helix of CUL4. The BPA-BPC double propeller creates a cavity that binds to substrate receptors by sandwiching an α-helix of these proteins. Although the majority of substrate receptors contain a WD40 domain, it has been shown that the WD40 domain primarily supports binding to DDB1, while it plays a more significant role in substrate recognition. The BPA-BPC double propeller of DDB1 and its associated substrate receptor proteins are oriented towards ROC1, positioning the substrate near the ROC1-bound E2 enzyme, thereby facilitating ubiquitylation.

Vertebrate genomes contain two *CUL4* genes—*CUL4A* and *CUL4B*—arising from gene duplication [16,17,18]. CUL4B contains a nuclear localization signal (NLS) in its N-terminus, which is absent in CUL4A, resulting in CUL4A localizing mainly in the cytosol, while CUL4B is primarily nuclear [19,20]. Since the NLS of CUL4B is located outside the binding region for DDB1 and ROC1, it does not interfere with CRL4 complex formation. The ancestral function of CUL4 is primarily related to chromatin regulation, as discussed below. This suggests that gene duplication allowed CUL4A to migrate from the nucleus to target cytosolic proteins, while CUL4B retains the ancient chromatin regulatory functions within the nucleus.

## 3. Organismal Phenotypes of *CUL4* Mutation

The organismal function of CUL4 has been investigated through gene disruption or suppression, as well as overexpression, in various model organisms, which are briefly summarized in this section (Table 1).

The disruption of *Cul4* in the yeast species *Schizosaccharomyces pombe* (*S. pombe*) and *Neurospora crassa* (*N. crassa*), along with the functional homolog Cul8 (also known as Rtt101) in *Saccharomyces cerevisiae* (*S. cerevisiae*), results in hypersensitivity to DNA damage, mitotic defects, and growth retardation [37,38,39,40,41,42]. Biochemical data discussed later support the notion that these defects are, at least in part, caused by the dysregulation of heterochromatin-mediated gene silencing, chromosome segregation, DNA replication, and DNA repair, all of which point to chromatin abnormalities.

The inactivation of *Cul-4* in a free-living transparent nematode *Caenorhabditis elegans* (*C. elegans*) causes developmental arrest at the second-instar larval stage and is accompanied by the enlargement of blast cell lineages with extensive DNA re-replication and accumulation [43]. The germline cells of such worms also show defects in the process of meiosis [28].

*Cul-4* disruption in a fruit fly *Drosophila melanogaster* (*D. melanogaster*) also leads to developmental arrest at the larval stage [44,45]. Wing-disk cells in *Cul-4* mutant flies exhibited reduced DNA replication, highlighting the essential role of Cul-4 in DNA replication. In contrast, during the focal DNA replication stage in ovarian follicle cells, Cul-4 depletion resulted in ectopic DNA replication. These findings suggest the dual functions of Cul-4 in DNA replication—facilitating the G1/S transition and suppressing abnormal DNA replication during the S phase. The loss of *Cul-4* function in the retina disrupts retinal development by downregulating the expression of retinal determination genes, while ectopically inducing the expression of negative regulators of eye development [46]. These findings suggest that Cul-4 plays a crucial role in cell differentiation through gene regulatory mechanisms.

The disruption of *CUL4A* in *Danio rerio* (zebrafish) embryos led to developmental defects in the heart and pectoral fins with the abnormal upregulation of *Tbx5a* gene [47]. Interestingly, human CUL4A and CUL4B both rescued the phenotype caused by CUL4A reduction, while zebrafish CUL4B could not, suggesting that human CUL4B and zebrafish CUL4B have divergent functions, likely involving different substrates. Zhao et al. also noted that CUL4B morpholino caused no apparent phenotype, indicating that CUL4A and CUL4B play distinct roles in development.

The first *Cul4a* knockout mice generated were mistakenly double knockouts of both the *Cul4a* gene and the adjacent *Pcid2* gene, which resides on the complementary strand of CUL4A [57]. Subsequently, two *Cul4a* knockout mouse models were developed, each targeting either the ROC1- [48] or DDB1-binding domains [50], and both exhibited no gross developmental abnormalities throughout their lifespans. However, further analyses revealed cardiac hypertrophy [49] and spermatogenic defects in male mice [51,58]. Additionally, drug-induced K14^+^ skin cell-specific Cul4a knockout mice were found to be resistant to UV irradiation-induced skin tumor formation originating from K14^+^ cells [48]. Enhanced global genome repair via the nucleotide excision repair (GG-NER) pathway plays a significant role in this protection.

The deletion of *CUL4B* in mice results in embryonic lethality, which is characterized by massive apoptosis in extra-embryonic tissues [52,53]. Studies using extra-embryonic tissue-derived XEN cells indicated that impaired proliferation is a likely mechanism behind the embryonic lethality. The disruption of *CUL4B* in the epiblast, but not in extra-embryonic tissues, allows *CUL4B*-deficient embryos to survive, and the resulting viable mice display defects in spatial learning and memory, which are consistent with the fact that *CUL4B* loss-of-function mutations are frequently associated with X-linked intellectual disability in humans [54,55,56]. However, several phenotypes observed in patients, such as short stature, obesity, small testes, and macrocephaly, were not present in *CUL4B* null mice. Subsequent studies revealed the altered differentiation of neural stem cells into glial cells, which was driven by the abnormal upregulation of relevant transcription factors [47].

In humans, *CUL4A* has been reported to be amplified in several cancers, including breast cancer [59,60,61], hepatocellular carcinoma [62], squamous cell carcinoma of the esophagus [63], adrenocortical carcinoma [64], non-small cell lung cancer [65], osteosarcoma [66], pleural mesothelioma [67], childhood medulloblastoma [68], epithelial ovarian tumors [69], and prostate cancer [70]. The causal role of CUL4A overexpression in tumorigenesis was demonstrated in breast cancer cells, where *CUL4A* silencing led to reduced cell proliferation, colony formation, cell motility, and tumor development [71,72,73]. Consistent with the tumorigenic role of CUL4A, the conditional overexpression of CUL4A in mouse lungs induced pulmonary tumors [74].

*CUL4B* amplification has also been observed in colon cancer [75], as well as in cervix, lung, esophagus, and breast cancers [76]. In mouse models, CUL4B overexpression promoted both spontaneous and DEN-induced hepatocarcinogenesis, supporting the crucial role of CUL4B in tumorigenesis [77]. Given that several CRL4 substrates identified to date either directly or indirectly regulate cell proliferation, the molecular mechanisms by which CUL4 dysregulation drives tumorigenesis appear to be complex and context-dependent.

Overall, these findings suggest that CRL4 functions are closely associated with DNA-templated processes. In the following sections, the reported molecular functions of CRL4 in various model organisms are summarized, providing further evidence in support of these CRL4 functions.

## 4. CRL4 Functions in *Trypanosoma*

*Trypanosoma*, positioned at an early branch of eukaryotic evolution, serves as a valuable model to explore the primitive functions of CRLs [15] (Figure 1b). The proteome-wide identification of *T. brucei* CUL4 (TbCul-D) bound proteins revealed associations with DDB1, ROC1, two DCAF proteins, and several nuclear proteins. One of the identified DCAF proteins is specific to *Trypanosoma*, while the other has an ortholog in *Leishmania*. The elucidation of the molecular functions of these DCAFs is anticipated to shed light on the earliest functions of CRL4. The same study identified two additional DCAF proteins, which are putative orthologs of human nuclear DCAFs; DCAF7 and DCAF13. Whether these proteins act as substrate receptors for CRL4, and if so, whether their substrates are orthologs of human proteins known to be regulated by CRL4—such as DNA ligase 1 for CRL4^DCAF7^ and DNA-crosslinked TOP1 for CRL4^DCAF13^—remains to be investigated (see Section 8).

## 5. CRL4 Functions in Fungi

Fungal genomes generally contain at least three prototypical *Cullin* genes, which correspond to homologs of the mammalian *CUL1*, *CUL3*, and *CUL4* [13]. Notably, *S. cerevisiae* does not contain *CUL4* based on sequence conservation but expresses a unique Cullin, *Cul8* (also known as *Rtt101*), which forms a CRL4-like complex with Mms1 as a counterpart of DDB1 [78]. *Cul8* is now considered a functional homolog of *CUL4*. Similar to other organisms, CRL4/8 complexes play essential roles in chromatin regulation within these yeast species (Table 2).

In the filamentous fungus *N. crassa*, Cul4 is a component of the histone methyltransferase complex DCDC (DIM-5/-7/-9, Cul4, DDB complex), where DIM-5 catalyzes the trimethylation of lysine 9 on histone H3 (H3K9) [41,42,87]. In this organism, H3K9me3 is essential for DNA methylation, serving as a scaffold for the HP-1/DIM-2 complex, which contains cytosine methyltransferase activity in the DIM-2 subunit. The critical role of Cul4/DDB1 in the DCDC complex was demonstrated by the loss of H3K9me3 and DNA methylation following *CUL4* or *DDB1* deletion. DIM-9/DCAF26 is considered a DCAF substrate receptor protein, as it directly binds to DDB1/DIM-8 and contains a WD40 domain [42,87]. Additionally, DIM-9 was found to be stabilized by DDB1, a feature shared by some, but not all, DCAF proteins.

Interestingly, *Cul4* mutants lacking the C-terminus (which prevents binding to ROC1) or mutants with a point mutation at K863 (the neddylation site) were still able to support H3K9 and DNA methylation to the same extent as wild-type Cul4, indicating that ubiquitin ligase activity is dispensable for DCDC function [88]. Instead, Cul4’s ubiquitin ligase activity is required for DCDC-independent functions, such as DNA repair. However, the molecular mechanisms by which Cul4 contributes to DNA repair in this organism have not yet been explored.

Among the phenotypes induced by CUL4 deletion (Table 1), mitotic defects and growth retardation are at least partially attributable to DCDC dysfunction, as DCDC-deficient strains exhibit abnormal chromosome segregation and growth defects due to the disruption of centromeric heterochromatin marked by H3K9me3. Interestingly, only *Cul4* and *Dim-8* (the *DDB1* gene) mutants were hypersensitive to DNA damage, supporting a role for Cul4/DDB1 in DNA repair that is independent of DCDC.

Heterochromatin formation in *S. pombe* relies on a H3K9 methyltransferase complex called CLRC (Clr4 methyltransferase complex), which contains a CRL4-like module comprising Cul4 (also known as Pcu4), Rik1 (a distant DDB1 homolog), Pip1 (a ROC1 homolog), and the putative DCAF protein Dos1/Clr8/Raf1. In contrast to *N. crassa*, the ubiquitin ligase activity of CLRC in *S. pombe* is essential for heterochromatin formation [89,90]. A later study identified H3K14 as a critical substrate of ubiquitylation, which is required for subsequent H3K9 methylation [89]. CLRC also targets the H3K4 methyltransferase Set1, which counteracts heterochromatinization, as well as the H3K9 demethylases Lsd1/2 for degradation [80].

In addition to its role in CLRC, *S. pombe* Cul4 forms a typical CRL4 complex, as found in multicellular eukaryotes, utilizing Ddb1. Cdt2 was shown to act as the DCAF substrate receptor, targeting Epe1, Spd1, and Cdt1 for ubiquitylation and degradation. Epe1 prevents the spread of H3K9 methylation into euchromatin, thereby maintaining the euchromatin/heterochromatin boundary. Although Epe1 associates with the entire heterochromatin region, CRL4^Cdt2^-mediated ubiquitylation and degradation within heterochromatin restrict Epe1 to the heterochromatin boundary [81]. Spd1, an inhibitor of ribonucleotide reductase, which generates dNTPs, is also targeted for degradation by CRL4^Cdt2^ during the S phase and in response to DNA damage, meeting the increased demand for the dNTP pool [39,82,83,91].

Cdt1 degradation by CRL4^Cdt2^ is a conserved mechanism across fungi and mammals, ensuring that DNA is replicated only once per cell cycle and preventing DNA re-replication [84]. Together, these studies reveal that CRL4 in *S. pombe* plays dual roles: supporting DNA replication by degrading Spd1 and Cdt1, and maintaining heterochromatin integrity by localizing Epe1 at the euchromatin/heterochromatin boundary.

*Cul8* was initially identified through a genetic screen aimed at uncovering genes involved in Ty1 transposition [92] and was later found to repress the transcription of telomeric genes [40]. However, compared to other fungi, the molecular mechanisms of Cul8 related to transcription remain largely unexplored. In contrast, significant progress has been made in understanding Cul8’s role in DNA replication and repair. In this context, Cul8 operates downstream of Rtt109, which catalyzes H3K56 acetylation, a histone mark incorporated into chromatin to ensure genome stability during DNA replication and repair [93,94].

It is well established that newly synthesized histone H3-H4 pairs are first captured by the histone chaperone Asf1, on which Rtt109 acetylates H3K56 before the pair is transferred to other chaperones, such as Rtt106 and Caf-1, which mediate histone incorporation into replicated DNA. The Cul8^Mms22^ complex was shown to ubiquitylate K56-acetylated H3 at three C-terminal lysine residues, facilitating the transfer of histones from Asf1 to Rtt106 [85]. In parallel, CUL8^Mms22^ links Asf1 to Caf-1, aiding in histone passage independently of the H3 ubiquitylation mechanism described above [95].

Importantly, these mechanisms appear to be conserved in human cells, as the binding of histones to chaperones downstream of ASF1 and the deposition of newly synthesized H3 into chromatin are compromised in CUL4A- or DDB1-depleted human cultured cells [85]. This biochemical evidence supports the notion that Cul8-based ubiquitin ligase is a functional homolog of CRL4 in chromatin regulation. Beyond histone H3, Cul8 also ubiquitylates Spt16, a component of the histone chaperone FACT, to promote its localization to the replication origin through an as-yet-undefined mechanism [86]. Since FACT is a part of the large replication progression complex (replisome), which includes essential replicative helicase MCM proteins, the deletion of Rtt101 significantly reduces MCM at the replication origin, indicating that Cul8 facilitates replisome formation via FACT.

Moreover, CUL8^Mms22^ was found to associate with the replisome, where it ubiquitylates as-yet-unidentified proteins crosslinked to DNA, thus removing obstacles to DNA replication [96] and promoting the restart of stalled replication forks [97].

Similar to *A. thaliana*, yeasts also encode CSA orthologs (Csa in *N. crassa*, Rad28 in *S. cerevisiae*, and Ckn1 in *S. pombe*); however, their role as DCAFs for CRL4 has not been documented. Although mutants of *Rad28* or *Ckn1* do not display defects in TC-NER activity [98,99], in the context of a disrupted secondary repair pathway known as UVER, which is unique to yeasts, the *Ckn1* mutant exhibits compromised TC-NER function and increased sensitivity to UV radiation [99]. This suggests that yeast CSA orthologs are also involved in the TC-NER pathway.

Collectively, these findings indicate that CRL4 and its functional homologs play pivotal roles in heterochromatin formation, DNA replication, and repair. The disruption of these processes leads to hypersensitivity to DNA damage, mitotic failure, and ultimately, growth retardation.

## 6. CRL4 Functions in Worms

In this and the next sections, we will overview the chromatin-related substrates and the roles of CRL4 in two model organisms, *C. elegans* and *D. melanogaster*, as outlined in Table 3.

As previously mentioned, the inactivation of *Cul-4* in *C. elegans* leads to developmental arrest at the second-instar larval stage, which is characterized by the enlargement of blast cell lineages [43]. In wild-type worms, these cells rapidly degrade the DNA replication factor CDT-1 by CRL4^CDT-2^ once DNA replication commences during the S phase. However, this CDT-1 degradation is absent in *Cul-4*-depleted cells, resulting in massive DNA re-replication and accumulation. The loss of one allele of *CDT-1* suppresses this DNA re-replication phenotype, reinforcing the critical role of CDT-1 dysregulation in the DNA replication defects caused by *Cul-4* inactivation.

In addition to CDT-1, *C. elegans* CRL4 has been reported to target Pol η and SKN-1 for degradation, using CDT-2 and WDR-23 as substrate receptors, respectively. Pol η, a well-characterized member of the Y-family DNA polymerases, plays a key role in replicating damaged DNA but at the cost of lower fidelity [107]. In response to DNA damage, Pol η is degraded by CRL4^CDT2^, with this degradation being counteracted by the SUMOylation of Pol η by the GEI-17 SUMO ligase [100]. This degradation is speculated to be a mechanism for removing Pol η from chromatin after damaged DNA is replicated.

Beyond its roles in DNA replication and repair, CRL4 is also involved in gene expression regulation in *C. elegans*, as demonstrated by its regulation of SKN-1. SKN-1 is a transcription factor that acts as a master regulator of oxidative stress responses through gene expression [108]. A genome-wide RNAi screen aimed at identifying genes involved in oxidative stress-inducible gene expression revealed the critical roles of proteasome subunits DDB1 and WDR-23 in the oxidative stress response [101]. Further analyses showed that WDR-23 directly binds to DDB1 and SKN-1, with CRL4^WDR-23^ mediating the degradation of SKN-1 in intestinal nuclei. Oxidative stress activates p38 MAPK, which inhibits SKN-1 degradation by CRL4^WDR-23^, contributing to the inducible expression of SKN-1-regulated genes.

Interestingly, while this mechanism is conserved, it functions as a backup in mammals. The mammalian ortholog of SKN-1, NRF2, is predominantly degraded by the CRL3-KEAP1 complex [109]. However, in the absence of KEAP1—similar to *C. elegans*, which lacks a KEAP1 ortholog—WDR23 (also referred to as DCAF11)-associated CRL4 ubiquitylates NRF2, thereby maintaining NRF2 at low levels [110].

Meiosis is a specialized process in germ cells where genetic information is partially exchanged between the homologous regions of sister chromatids through recombination, prior to the formation of haploid cells [111]. The DCAF protein GAD-1 enables CRL4 to participate in this process, as evidenced by the fact that mutants of *cul-4*, *ddb-1*, *gad-1*, or a component of the CSN all exhibit defects in synaptonemal complex (SC) formation, which is essential for meiotic recombination [28]. Interestingly, *Rbx-1/2* mutants do not display this phenotype, suggesting that either ubiquitin ligase activity is not required at this stage or an atypical E2 enzyme is involved in germ cells. While it is suspected that increased expression and transposition of the *Tc3* transposable element, which causes DNA damage, may be partially responsible for SC misformation in CRL4^GAD-1^-inactivated germ cells, the precise mechanisms and substrates involved in this process remain unknown.

## 7. CRL4 Functions in Flies

Studies in *D. melanogaster* have also demonstrated the critical roles of CRL4 in both DNA replication and gene expression. Similar to what was observed in *C. elegans*, CRL4^L(2)dtl^ (the *Drosophila* homologous complex of CRL4^CDT2^) is responsible for the degradation of double parked (Dup), the *Drosophila* homolog of CDT1 [45]. Additionally, the E2f1 transcription factor, a key activator in the E2F family that drives cell cycle progression, is degraded by CRL4^L(2)dtl^ during the S phase [102]. This degradation is essential for normal cell cycle progression in the wing imaginal disks and for the endocycle in larval salivary glands [112].

CRL4 has also been reported to mediate the degradation of cell cycle regulators Dacapo (Dap), the *Drosophila* homolog of Cip/Kip family CDK inhibitor, and cyclin E, though the substrate adaptor proteins responsible for these actions and the physiological significance of these regulatory processes remain unclear [103].

CRL4 was found to regulate the light-induced degradation of Cry protein, with Ramshackle, the *Drosophila* homolog of BRWD3, serving as the DCAF substrate receptor [104]. Although Cry is not directly involved in gene expression, its regulation of the circadian clock protein Tim impacts gene expression changes through the Clock–Cycle heterodimer [113], suggesting that CRL4 might indirectly influence circadian gene expression.

Recently, it was demonstrated that CRL4^BRWD3^ regulates histone H3 lysine 4 (H3K4) methylation levels, a modification crucial for transcriptional regulation [114]. H3K4 can exist in mono-, di-, and trimethylated states (designated as H3K4me1/2/3, respectively), with these levels being dynamically regulated by the opposing activities of methyltransferases and demethylases. Specifically, BRWD3-associated CRL4 ubiquitylates KDM5 (also known as Lid), an H3K4me3/2 demethylase, targeting it for degradation, which subsequently increases H3K4me3 levels. Although H3K4me3 is extensively associated with transcriptional activity, the authors did not observe a direct correlation with gene expression in the differentially expressed genes, i.e., upregulated genes did not necessarily exhibit increased H3K4me3, and this was similarly observed for downregulated genes. This may be attributed to the transient RNAi-based knockdown of CUL4 and BRWD3, which may require more time to manifest in H3K4 methylation’s impact on gene expression. Whether this regulatory mechanism is conserved across other organisms remains to be investigated.

*D. melanogaster* is frequently used to investigate the mechanisms of cell competition, a process where unfit cells are removed by their neighboring cells. The involvement of Mahjong (the *Drosophila* homolog of DCAF1) in this process has been suggested [115], and a recent report identified the Xrp1 transcription factor as a critical substrate of CRL4^DCAF1^. The degradation of Xrp1 by CRL4^DCAF1^ leads to the transcriptional repression of genes associated with cell elimination, thereby conferring resistance to competition [105].

Cell competition is not the only outcome of cell–cell interactions; most non-transformed cells exhibit contact-dependent inhibition of proliferation through the activation of the Hippo pathway [116,117]. *D. melanogaster* is also commonly used in studies of the Hippo pathway, which regulates organ size [118]. The activation of the Hippo pathway by cell contact triggers a kinase cascade, leading to the nuclear exclusion and inactivation of Yki, a transcription factor that promotes cell proliferation. DCAF12 has been reported to recognize Yki and recruit CRL4, resulting in Yki’s ubiquitination and subsequent degradation via the proteasome in the nucleus, preventing Yki hyperactivation [106]. This mechanism has also been shown to be conserved in mammalian cells [106].

Taken together, these findings highlight the significant role of CRL4 in regulating DNA replication, repair, and gene expression in both *C. elegans* and *D. melanogaster*.

## 8. CRL4 Functions in Mammals

Although *CUL4* was first identified in worms [119], most of the knowledge about CRL4 function has been accumulated through studies in mammalian cells, leading to the availability of many excellent review articles on this topic.

In this section, we first summarize the literature on DNA and histone modification-related proteins that are modulated by CRL4, using these examples to illustrate CRL4-mediated chromatin regulation. We then provide a brief overview of CRL4 functions in relation to nucleus-localizing DCAF proteins [120,121]. The chromatin-related substrates of CRL4 highlighted in this section are summarized in Table 4. Readers are encouraged to refer to the relevant review articles for further details, where available.

### 8.1. DNA Modification-Related Proteins

DNA methylation at the C-5 position of cytosine (5mC) is a crucial chromatin modification that regulates gene expression. There are three DNA methyltransferases in humans: DNA methyltransferase 3A and 3B (DNMT3A/B), which catalyze de novo DNA methylation, and DNMT1, which acts as a maintenance methyltransferase during cell division, targeting hemimethylated regions in the S phase [165]. DNMT3A, in particular, plays a crucial tumor suppressor role in the hematopoietic system and is frequently mutated in clonal hematopoiesis (CH). Approximately two-thirds of missense mutations and several 3 bp in-frame deletions in the *DNMT3A* gene cause instability of the DNMT3A protein, with CRL4B^DCAF8^ identified as the ubiquitin ligase that ubiquitylates and degrades mutant DNMT3A [122]. These findings suggest that DCAF8 monitors abnormal DNMT3A for degradation, contributing to disease pathogenesis, and provide a rationale for developing DCAF8 inhibitors as potential treatments for DNMT3A mutation-induced CH. Notably, wild-type DNMT3A also binds to DCAF8, indicating that CRL4B^DCAF8^ also regulates wild-type DNMT3A.

Interestingly, DNMT3B has been shown to interact with NEDDylated CUL4A, and NEDD8 enhances DNMT3B-dependent DNA methylation [166]. This suggests that DNMT3B is also regulated by CRL4, though the precise mechanisms remain unclear.

DNMT1 stability is regulated by methylation-dependent ubiquitylation. Outside the S phase, when hemimethylated DNA is absent, SET7 methylates a specific lysine residue on DNMT1, which is recognized by the L3MBTL3 methyl-reader protein, recruiting CRL4^DCAF5^ for ubiquitylation and degradation of DNMT1. During the S phase, LSD1 removes this methyl group, stabilizing DNMT1 [123].

In the G2/M phase, methylated DNA in centromeric and pericentromeric regions is maintained by the ATP-dependent chromatin remodeler LSH (also known as HELLS or PASG) [167]. CRL4^DCAF8^, in addition to targeting DNMT3B, also degrades LSH. In response to oxidative stress, CRL4^DCAF8^ ubiquitylates and degrades LSH, leading to DNA demethylation and the upregulation of genes, particularly those related to ferroptosis [124]. Although the precise mechanisms by which oxidative stress triggers CRL4^DCAF8^-mediated LSH ubiquitylation and the specificity of target genes in this pathway remain unclear, forced DNA demethylation through TET2 catalytic domain overexpression attenuates the interaction between DCAF8 and LSH. This suggests that methylated DNA, maintained by LSH, promotes LSH degradation, contributing to the dynamic regulation of DNA methylation.

The effects of 5mC are mediated by methyl DNA-binding domain proteins (MDBs), which recruit histone-modifying enzymes and chromatin remodelers to exert their repressive functions [168]. Among these, MeCP2 has been recently shown to be ubiquitinated and degraded by CRL4^DCAF13^ in the ovary [125]. In secondary ovarian follicles, DCAF13 is upregulated, which is correlated with the reduction in MeCP2 levels, promoting gene expression and follicle growth. Given that CRL4^DCAF13^ regulation of MeCP2 has been demonstrated in HeLa cells, it is plausible that this mechanism functions in other tissues as well.

A breakthrough in the DNA methylation field was the discovery of TET dioxygenases, which catalyze the iterative oxidation of 5mC, ultimately leading to DNA demethylation, demonstrating that DNA methylation is reversible [169]. TET proteins are monoubiquitylated at highly conserved lysine residues by CRL4A/B^DCAF1^ [126]. This monoubiquitylation enhances the binding of TET proteins to DNA, which is essential for their catalytic function in cells. However, the detailed mechanism by which monoubiquitylation promotes TET binding to DNA remains unclear.

The final step of DNA demethylation is catalyzed by the base excision repair enzyme TDG (thymine DNA glycosylase), which is also targeted by CRL4, with CDT2 as the substrate receptor. Similar to other CRL4A/B^CDT2^ substrates, TDG degradation occurs during the S phase and in response to UV damage [127,128]. Although the physiological function of this regulated degradation is still unclear, it suggests that DNA demethylation is suppressed during the S phase or following UV irradiation.

The terminal regions of eukaryotic genomic DNA are structured into specialized complexes known as telomeres, which progressively shorten with aging [170]. Telomere length is re-established by a Zscan4-dependent mechanism during early embryogenesis. Zscan4 expression is tightly regulated, with KAP-1 acting as a repressor. CRL4-DCAF11 has been identified to ubiquitinate KAP-1, targeting it for degradation and thereby maintaining optimal levels of Zscan4 expression [129]. CRL4^DCAF11^-mediated Zscan4 expression is evident in embryos and embryonic stem cells, but not in differentiated fibroblasts, indicating the cell-type-specific effects of CRL4^DCAF11^.

### 8.2. Histone Modification-Related Enzymes

It is well established that two copies each of the histones H2A, H2B, H3, and H4 form the nucleosome core particle, around which 146 base pairs of DNA are wrapped [171]. In vertebrate cells, it is estimated that 5–15% of H2A and 1–2% of H2B are conjugated with ubiquitin, making them among the most abundant ubiquitylated proteins in the nucleus [172]. Specifically, monoubiquitylation on K119 of H2A and K120 of H2B are well characterized for their respective roles in gene repression and gene activation [7,173].

CRL4B contributes to H2A K119 ubiquitylation via DCAF proteins RBBP4/7 in vivo and in vitro [76]. Hu et al. demonstrated that CUL4B binds to tumor-related gene promoters and negatively regulates their expression by catalyzing H2A K119 ubiquitylation, in collaboration with PRC2 (Polycomb Repressive Complex 2), which catalyzes the repressive histone mark H3K27 methylation. Further analyses revealed that CUL4B also associates with the H3K9 methyltransferase SUV39H1 and DNMT3A, both involved in gene repression through histone and DNA modifications [174]. This suggests that CUL4B may resemble the ancestral CUL4 more closely than CUL4A, given CUL4’s involvement in H3K9 methylation and DNA methylation in single-cell eukaryotes.

Other CRL4-based ubiquitin ligases, such as CRL4A/B^DDB2^, have been reported to ubiquitylate H2A in vitro [21,22,130], though it remains unclear whether this occurs in cells. In contrast, the CRL4A/B^DDB2^-mediated ubiquitylation of H3 and H4 has been demonstrated both in vitro and in vivo in response to UV damage [130]. The transient nature of H3/H4 ubiquitylation following UV exposure, along with the observation that a fraction of ubiquitylated H3 is released from chromatin, suggests that this process may loosen chromatin structure to facilitate DNA repair. This study implied that degradation may not be the sole fate of CRL4-ubiquitylated proteins. However, the ubiquitylation-coupled degradation of H4 has been observed, requiring prior methylation on the 24th residue (H4D24me), which is recognized by the DCAF protein DCAF1, leading to the degradation of damaged H4 [131].

Histone variant proteins, such as CENP-A, can replace canonical histones. CENP-A specifically localizes to centromeres and alters chromatin structure to facilitate kinetochore formation [175]. The monoubiquitylation of CENP-A by CRL4A^COPS8^ is required for its association with the chromatin assembly factor HJURP and subsequent loading onto the centromere [132]. Another CUL4 complex, CRL4A^RBBP4/7^, is also essential for CENP-A loading, suggesting multiple points where CRL4 is involved in CENP-A incorporation into chromatin [133].

Replicated DNA is assembled into chromatin, necessitating the supply of new histone proteins. To meet this demand, SLBP binds to histone mRNAs, promoting their translation during the S phase. In the late S phase, SLBP is degraded through CRL4^DCAF11^-mediated ubiquitylation to prevent the overproduction of histones [134]. In the context of histone incorporation during the S phase, CRL4A/B^DCAF14^ was recently discovered to ubiquitylate the histone chaperone protein SPT16, facilitating histone release, which is critical for histone incorporation into newly synthesized DNA [120]. These findings underscore CRL4’s extensive role in histone regulation and chromatin remodeling.

Histone modifications extend beyond ubiquitylation, with other modifications such as acetylation, methylation, and phosphorylation being commonly observed. CRL4 regulates DNA replication and gene expression by targeting several histone-modifying enzymes. One key example is the histone acetyltransferase HBO1 (also known as KAT7 and MYST2), which acetylates histones H3 and H4 at DNA replication origins. HBO1 functions as a DNA replication factor, but upon DNA damage, it is rapidly phosphorylated by ATM and ATR kinases and subsequently degraded by CRL4A/B^DDB2^ [135]. This degradation is proposed to prevent the initiation of DNA replication on damaged DNA, thus maintaining genome integrity.

Another important CRL4 substrate is the H4K20 methyltransferase SET8 (also known as PR-SET7 and KMT5a), which plays critical roles in genome integrity, DNA replication, chromosome condensation, and gene transcription [176]. Multiple studies have reported that SET8 is degraded by CRL4A/B^CDT2^ during the S phase and in response to UV irradiation, similarly to CDT1 [136,137,138,139,140]. The disruption of CRL4A/B^CDT2^-mediated SET8 degradation results in defects such as DNA re-replication, abnormal chromatin compaction, and transcriptional repression, underscoring the critical role of CRL4 in these processes through the regulation of SET8 and the subsequent reduction in H4K20 methylation.

SUV39H1/2 catalyzes H3K9 methylation to generate H3K9me3, a repressive histone modification that inhibits transcription. In the investigation of oocytes, the Heng-Yu Fan group discovered that DCAF13, the same protein targeting MeCP2 in the ovary as described above, is responsible for the ubiquitylation and degradation of SUV39H1 in zygote, facilitating the removal of H3K9me3 to support zygotic gene activation [141].

H3K27me3 is an established repressive histone mark, which is written by the EZH2 subunit of the PRC2 complex as previously described, while two proteins serve as erasers: KDM6A and KDM6B (also known as UTX and JMJD3, respectively). A recent study revealed that the CRL4B^COP1^ complex ubiquitylates KDM6A, targeting it for degradation [142]. Intriguingly, USP7 was shown to stabilize KDM6B via deubiquitylation [177], and the same study identified DCAF7 (also known as WDR68 or Han11) as a binding partner of KDM6B, suggesting the untested hypothesis that CRL4^DCAF7^ may ubiquitylate KDM6B for degradation.

Menin is an essential component of the KMT2A and KMT2B H3K4 methyltransferase complexes [178]. In leukemia, particularly in acute myeloid leukemia (AML) and acute lymphoblastic leukemia (ALL), the *KMT2A* (also known as *MLL)* gene undergoes translocations with a variety of partner genes, leading to the formation of oncogenic KMT2A fusion proteins [179]. In such a complex, menin interacts with the H3K79 histone methyltransferase DOT1L [180,181] and recruits DOT1L to KMT2A target genes, leading to increased H3K79 methylation and transcriptional upregulation, which is crucial for the leukemic transformation of myeloid progenitors caused by KMT2A mutations [182]. Recently, DCAF7 was identified as a binding partner of menin, facilitating its ubiquitylation and degradation by CRL4B [143], presenting a potential therapeutic target for malignancies driven by menin.

WDR5, an essential component of all H3K4 methyltransferases (KMT2A, B, C, D, F and G), plays a key role in promoting gene expression. It was previously reported that CRL4B targets WDR5 for degradation in the nucleus, without the involvement of DCAF substrate receptors [20,144]. The knockdown of CUL4B leads to the accumulation of WDR5 and increased H3K4 trimethylation at several gene promoters, resulting in enhanced gene expression. This illustrates CRL4B’s regulatory role in gene expression by modulating histone methylation levels.

In conclusion, these findings support the notion that CRL4 is a critical regulator of chromatin structure and function in mammalian cells, targeting both histones and DNA through mechanisms that influence key processes such as DNA methylation and gene expression.

### 8.3. Other Substrates Targeted by Nuclear DCAFs

In addition to the previously mentioned substrates, CRL4^DCAF1^ also ubiquitylates transcription factors (FoxM1, MyoD, p53, RORa, TR4), a DNA replication factor (MCM10), DNA repair proteins (UNG2 and SMUG1), and a DNA recombinase (RAG-1) to regulate cell cycle progression and differentiation, as summarized in review articles [183,184]. Notably, DCAF1 recognizes methylated RORa, exemplifying methyl degron-dependent degradation, similar to the function of DCAF5 [148]. However, it remains unclear whether other substrates also require acetylation. A key feature of DCAF1 is its exploitation by several viral proteins to promote cellular protein degradation, aiding viral survival, proliferation, and dissemination. The current insights into this intriguing research area are further outlined in the recent review articles [185,186].

In addition to DNMT1, CRL4^DCAF5^ also recognizes methyl degrons on E2F1, SMARCC1/2, and SOX2 via L3MBTL3, facilitating their ubiquitylation and subsequent proteasome-mediated degradation [123,154]. This subject is comprehensively reviewed in a recent article [187].

CRL4^DCAF7^, in addition to targeting menin for degradation as previously mentioned, also ubiquitylates and degrades DNA ligase I (LIG1), which is responsible for connecting Okazaki fragments during lagging strand DNA synthesis. This occurs under conditions of proliferation inhibition when its substrate (DNA fragment) is absent [155].

DCAF13 plays a role in DNA repair by targeting TOP1 DNA–protein crosslinks (TOP1-DPCs) for degradation [156]. Consequently, the combination therapy involving TOP1 and NEDDylation inhibitors has shown effectiveness in treating colorectal and potentially other types of cancers.

The most deleterious form of DNA damage is the double-strand break (DSB), which is recognized by Ku heterodimers, subsequently leading to the recruitment of DNA-PKcs to initiate the DNA repair process [188]. CRL4A^CDT2^ ubiquitylates DNA-PKcs for degradation, rendering cells vulnerable to DSBs [157].

In addition to KDM6a, CRL4^COP1^ targets several transcription factors, including C/EBPβ, c-JUN, ETV5, and ETS1, 2, as well as p53, for degradation, thereby modulating gene expression [158,159,160,161,162,163].

Light encompasses a spectrum of different wavelengths, among which UV light can cause DNA damage by inducing the formation of cyclobutane pyrimidine dimers (CPDs) and pyrimidine (6-4) pyrimidone photoproducts (6,4-PPs) [189]. CPDs and 6,4-PPs obstruct and stall RNA Pol II progression in actively transcribed genes. In response, CSB recognizes the stalled Pol II and recruits DNA repair-related proteins and then CSA, a DCAF protein, ultimately leading to the degradation of CSB to complete TC-NER and facilitate the subsequent recovery of transcription [164].

## 9. Concluding Remarks

The current body of knowledge highlights the conserved and specialized functions of CRL4. To facilitate further comparative analysis of CRL4’s role in eukaryotic cells from an evolutionary perspective, the presence of orthologs of representative mammalian DCAFs discussed in this review in various eukaryotic organisms is summarized in Table 5.

One of the most conserved DCAFs is CSA, which plays a critical role in DNA repair at mRNA-producing loci (Figure 3a). This underscores the necessity for eukaryotic cells to possess specialized mechanisms to safeguard Pol II-transcribed regions from DNA damage, likely because such damage could lead to mRNA information alterations and the subsequent disruption of protein homeostasis. The functional ortholog of CSA in trypanosomes is yet to be identified, and its discovery would provide valuable insights into the evolutionary trajectory of CRL4 function.

Another NER pathway, GG-NER, is initiated by damage recognition through DDB2. Previously, it was proposed that DDB2 is only found in mammals [197], but it has since been discovered in plants as well. However, further research is needed to elucidate how the function of DDB2 is executed in species that lack this protein and whether CRL4 plays a role in these processes.

COP1 is evolutionarily conserved from yeast to mammals, though its partners and substrates have diverged [198]. In plants, COP1 forms a complex with SPA, a plant-specific protein, acting as a DCAF for CRL4 [199]. In contrast, in mammals, COP1 binds indirectly to DDB1 through DET1 [159]. The mechanisms driving these differences remain unclear, but studies in yeast and insects could provide insights into the processes underlying these divergences, potentially explaining the substrate diversification.

One of the earliest and most well-characterized functions of CRL4 is the prevention of DNA re-replication through the degradation of CDT1, with CDT2 (also known as DCAF2) serving as the DCAF (Figure 3b). This function has been clearly demonstrated in yeast, insects, and mammals, but plants appear to lack CDT2. However, since CDT1 activity is regulated by multiple pathways [200,201], these alternative mechanisms likely compensate for the absence of CDT2 in plants.

Other DCAFs present in yeast, insects, and mammals, but absent in plants, include DCAF11 and DCAF13. DCAF13 is one of four DCAFs identified in trypanosomes, suggesting its early emergence in eukaryotic evolution. Since CRL4^DCAF13^ has been predominantly studied in mammalian systems, it remains to be determined whether its functions can be traced back to ancient eukaryotic species. In comparison, the data on DCAF11 remain comparatively sparse, with the primary focus on its regulation of SKN-1/NRF2 (Figure 3c), underscoring the need for additional research to elucidate the functions of this evolutionarily conserved DCAF.

The intriguing case of DCAF1, which appears to be present in plants, insects, and mammals, but absent in trypanosomes and yeasts, suggests that it is specific to multicellular eukaryotes. This is consistent with its demonstrated role in cell–cell communication, though many additional functions, particularly those related to transcription, have been uncovered. Thus, it is important to explore the potential links between its transcriptional regulatory roles and its functions in cell–cell or inter-organ communication. Furthermore, the fact that DCAF1 can be hijacked by several viruses highlights the potential for further research on this DCAF to broaden our understanding of CRL4.

With the rapidly expanding field of targeted protein degradation (TPD) [202,203], spurred by the discovery of thalidomide-induced neo-substrates of CRL4 [204], a deeper understanding of CRL4 is now more critical than ever. Notably, CRL4 substrates are not always directed towards degradation [130], suggesting that ubiquitylated neo-substrates targeted by CRL4-based TPD may exhibit unintended secondary functions. We believe it is crucial to investigate the regulatory mechanisms governing the ubiquitylation of CRL4 substrates in greater detail. For instance, it is important to determine whether the specific structure of the substrate, substrate-specific deubiquitylases, or CRL4 itself influences the type and length of the ubiquitin chain. Although this review focuses on chromatin, the remarkable functions of CRL4 extend beyond chromatin-related processes. To achieve a comprehensive evolutionary perspective on CRL4, the elucidation of its non-chromatin and cytosolic functions is indispensable. For instance, numerous studies have demonstrated the involvement of CRL4 in autophagy through the ubiquitylation of key substrates, including Beclin 1 [205], DCAF3/AMBRA1 [206], WIPI2 [207], MAGE-A3/6 [208], and LAMP2 [209], thereby establishing a connection between the two protein degradation systems. Coordinated research efforts, involving scientists from diverse fields, are essential to uncover the full scope of CRL4’s roles and to harness its potential for therapeutic applications, not only in humans but also in various organisms, contributing to a healthier and more sustainable world.

## Figures and Tables

**Figure 1 cells-14-00063-f001:**
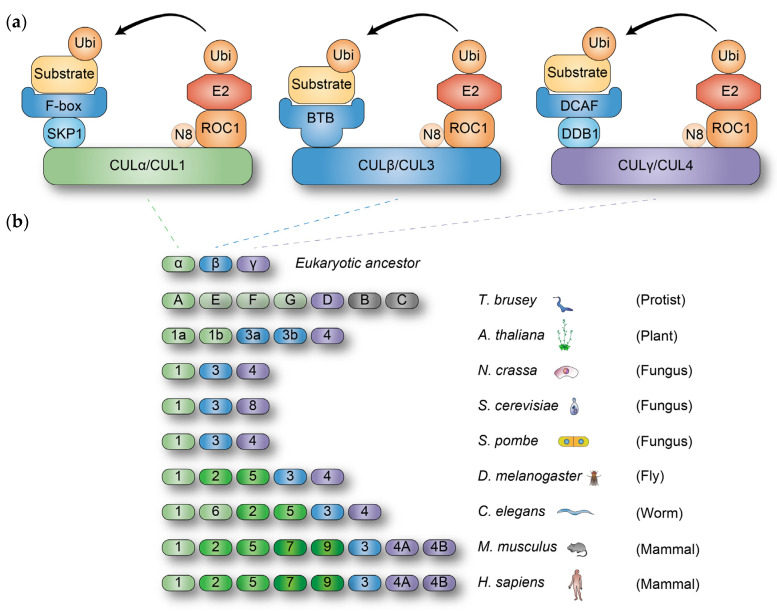
(**a**) The composition of CRLα/1, CRLβ/3, and CRLγ/4 ubiquitin ligases. Each Cullin protein depicted here associates with a ubiquitin-charged E2 enzyme through ROC1, functioning as a ubiquitylation module. For CRLα/1, SKP1 operates as a substrate adaptor, linking with F-box proteins to recruit substrate proteins. In contrast, CRLβ/3 functions without a substrate adaptor; instead, CRLβ/3 directly interacts with substrate receptor BTB proteins. CRLγ/4 employs DDB1 as a substrate adaptor, enabling the engagement of substrate receptor DCAF proteins. Every CUL protein is covalently attached to NEDD8 (N8) to form an active complex. (**b**) The evolution of Cullin proteins in eukaryotic cells. The common ancestor of eukaryotic cells is estimated to have contained three Cullin genes: *Culα*, *Culβ*, and *Culγ*. Through evolutionary divergence, Culα gave rise to several Cullins, depicted in green. Culβ served as the precursor for CUL3, which is represented in blue. CUL4 originated from the gene expansion of Culγ, which is illustrated in purple. Notably, mammalian cells possess two paralogs of CUL4, designated as CUL4A and CUL4B.

**Figure 2 cells-14-00063-f002:**
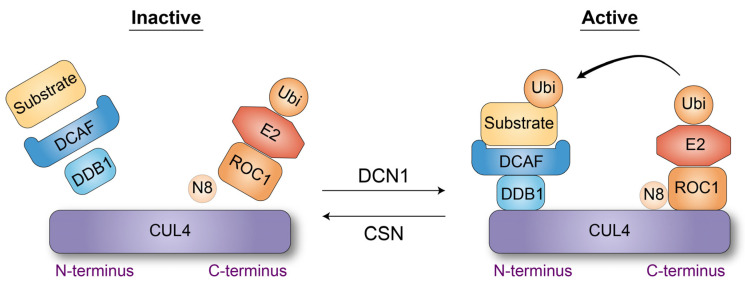
The structure and regulation of the CRL4 ubiquitin ligase. In its inactive state, NEDD8 (N8) is removed by the COP9 signalosome (CSN), thereby disrupting the substrate recognition module (DDB1/DCAF1/substrate) and the ubiquitylation module (ROC1/E2/ubiquitin). DCN1, in coordination with ROC1, facilitates the conjugation of N8 to a specific lysine residue on CUL4, resulting in the assembly of the active CRL4 ubiquitin ligase complex. The substrate recognition module associates with the N-terminal region of CUL4, while the ubiquitylation module interacts with the C-terminal region of CUL4.

**Figure 3 cells-14-00063-f003:**
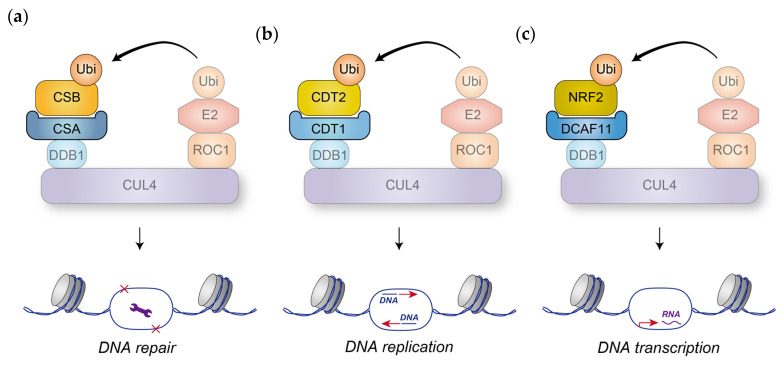
Evolutionarily conserved DCAFs and their established functions in mammalian cells. (**a**) CRL4 facilitates the ubiquitylation and subsequent proteasomal degradation of CSB via the DCAF protein CSA, playing a crucial role in the DNA repair process, particularly transcription-coupled nucleotide excision repair (TC-NER). While it remains uncertain whether CSA similarly recruits CSB to CRL4 in yeast and plants, organisms harboring *CSA* gene mutations exhibit impaired TC-NER and heightened sensitivity to UV-induced damage, implying a conserved role for CSA across species. (**b**) CDT2 mediates the recruitment of CDT1 to CRL4, facilitating its ubiquitylation and subsequent proteasomal degradation in yeast, insects, and mammals. The apparent absence of CDT2 in plants suggests the existence of alternative dominant mechanisms for CDT1 regulation, rendering CRL4^CDT2^ non-essential in plant systems. (**c**) In worms and mammals, DCAF11-associated CRL4 mediates the ubiquitylation of NRF2, a transcription factor critical under stress conditions, thereby maintaining NRF2 at low levels under normal circumstances. However, in mammalian cells, CRL3^KEAP1^ primarily facilitates NRF2 degradation, positioning CRL4^DCAF11^ as a secondary or backup regulatory mechanism.

**Table 1 cells-14-00063-t001:** Phenotypes of loss-of-function *CUL4* mutations.

Species	Cul4 Gene	Phenotype	References
*S. cerevisiae* *S. Pombe* *N. crassa*	*Cul8/Cul4*	Hypersensitivity to DNA damage	[37,38,39,40,41,42]
		Growth retardation	
		Mitotic defect	
*C. elegans*	*Cul-4*	Developmental defects	[28,43]
		DNA re-replication	
*D. melanogaster*	*Cul-4*	Developmental defects	[44,45,46]
*D. rerio*	*Cul4a*	Developmental defects	[47]
*M. musculus*	*Cul4a*	Resistance to skin cancer	[48]
		Cardiac hypertrophy	[49]
		Male infertility	[50,51]
	*Cul4b*	Lethality at embryonic stage	[52,53]
*H. sapiens*	*CUL4B*	X-linked intellectual disability	[54,55,56]

**Table 2 cells-14-00063-t002:** Chromatin-related CRL4 substrates in yeasts.

Species	Substrate	Receptor	Function of CRL4	References
*S. pombe*	Histone H3	Dos1	Promotion of H3K9 methylation	[79]
	Lsd1/2	Dos1	Inhibition of H3K9 demethylation	[80]
	Set1	Dos1	Inhibition of H3K4 methylation	[80]
	Epe1	Cdt1	Establishment of heterochromatin boundary	[81]
	Spd1	Cdt1	Promotion of dNTP synthesis	[39,82,83]
	Cdt2	Cdt1	Inhibition of DNA re-replication	[84]
*S. cerevisiae*	Histone H3	Mms22	Promotion of histone transfer to replicated DNA	[85]
	Spt16	Mms22	Promotion of DNA replication	[86]

**Table 3 cells-14-00063-t003:** Chromatin-related CRL4 substrates in insects.

Species	Substrate	Receptor	Function of CRL4	Reference
*C. elegans*	CDT-1	CDT-2	Inhibition of DNA re-replication	[43]
	Pol η	CDT-2	Completion of DNA repair	[100]
	SKN-1	WDR-23	Inhibition of stress-related gene expression	[101]
*D. melanogaste*	Dup (Cdt1)	L(2)dtl	Inhibition of DNA re-replication	[45]
	E2f1	L(2)dtl	Promotion of cell proliferation	[102]
	Dap	Unknown	Promotion of cell proliferation	[103]
	Cyclin E	Unknown	Inhibition of cell proliferation	[103]
	Cry	Ramshackle (Brwd3)	Inhibition of circadian gene expression	[104]
	Xrp1	Mahjong (Dcaf1)	Inhibition of cell elimination-related gene expression	[105]
	Yki	Dcaf12	Inhibition of cell proliferation-related gene expression	[106]

**Table 4 cells-14-00063-t004:** Chromatin-related CRL4 substrates in mammals.

Category	Substrate	Receptor	Function of CRL4	References
DNA modification-related	DNMT3A	DCAF8	Inhibition of DNA methylation	[122]
	DNMT1	DCAF5/L3MBTL3	Inhibition of DNA methylation	[123]
	LSH	DCAF8	Promotion of DNA demethylation	[124]
	MeCP2	DCAF13	Promotion of gene expression	[125]
	TET1/2/3	DCAF1	Promotion of DNA demethylation	[126]
	TDG	CDT2	Inhibition of DNA demethylation	[127,128]
	KAP1	DCAF11	Promotion of gene expression	[129]
Histone modification-related	Histone H2A	RBBP4/7	Inhibition of gene expression	[76]
	Histone H2A	DDB2	Promotion of DNA repair	[21,22,130]
	Histone H3	DDB2	Promotion of DNA repair	[130]
	Histone H4	DDB2	Promotion of DNA repair	[130]
	Histone H4	DCAF1	Promotion of DNA repair	[131]
	CENP-A	COPS8	Promotion of centromere formation	[132]
	CENP-A	RBBP4/7	Promotion of centromere formation	[133]
	SLBP	DCAF11	Inhibition of histone translation	[134]
	SPT16	DCAF14	Promotion of DNA replication	[120]
	HBO1	DDB2	Inhibition of DNA replication	[135]
	SET8	CDT2	Inhibition of DNA re-replication and transcription	[136,137,138,139,140]
	SUV39H1	DCAF13	Promotion of gene expression	[141]
	KDM6A	COP1	Promotion of gene expression	[142]
	Menin	DCAF7	Inhibition of gene expression	[143]
	WDR5	Not required	Inhibition of gene expression	[20,144]
Others	NRF2	DCAF11	Inhibition of gene expression	[110]
	FoxM1	DCAF1	Inhibition of gene expression	[145]
	MyoD	DCAF1	Inhibition of gene expression	[146]
	p53	DCAF1	Inhibition of gene expression	[147]
	RORa	DCAF1	Inhibition of gene expression	[148]
	TR4	DCAF1	Inhibition of gene expression	[149]
	MCM10	DCAF1	Inhibition of DNA replication	[150]
	UNG2	DCAF1	Inhibition of DNA repair	[151]
	SMUG1	DCAF1	Inhibition of DNA repair	[151]
	RAG1	DCAF1	Inhibition of DNA recombination	[152]
	E2F1	DCAF5/L3MBTL3	Inhibition of gene expression	[123]
	DNMT1	DCAF5/L3MBTL3	Inhibition of DNA methylation	[123]
	SMARCC1	DCAF5/L3MBTL3	Inhibition of gene expression	[153]
	SMARCC2	DCAF5/L3MBTL3	Inhibition of gene expression	[153]
	SOX2	DCAF5/L3MBTL3	Inhibition of gene expression	[154]
	LIG1	DCAF7	Inhibition of DNA replication	[155]
	TOP1-DPCs	DCAF13	Promotion of DNA repair	[156]
	DNA-PKcs	CDT2	Inhibition of DNA repair	[157]
	C/EBPβ	COP1	Inhibition of gene expression	[158]
	c-JUN	COP1	Inhibition of gene expression	[159]
	ETV5	COP1	Inhibition of gene expression	[160]
	ETS1	COP1	Inhibition of gene expression	[161]
	ETS2	COP1	Inhibition of gene expression	[161]
	p53	COP1	Inhibition of gene expression	[162,163]
	CSB	CSA	Completion of DNA repair	[164]

**Table 5 cells-14-00063-t005:** Conservation of representative nuclear DCAFs. USCS genome browser [190], HMMER web server [191], Saccharomyces Genome Database [192], PomBase [193], The Arabidopsis Information Resource [194], WormBase [195], and FlyBase [196] were searched for finding DCAF orthologs. A check mark denotes the presence, while a minus sign in parentheses indicates the absence of the corresponding DCAFs.

Mammalian DCAF	Trypanosoma	Yeasts	Plants	Insects
DCAF1	(-)	(-)	✔	✔
CDT2 (DCAF2)	(-)	✔	(-)	✔
DCAF5	(-)	(-)	(-)	(-)
DCAF7	(-)	✔	(-)	(-)
DCAF8	(-)	(-)	(-)	(-)
DCAF11	(-)	✔	(-)	✔
DCAF12	(-)	(-)	(-)	✔
DCAF13	✔	✔	(-)	✔
DCAF14	(-)	(-)	(-)	✔
COP1	(-)	✔	✔	✔
CSA	(-)	✔	✔	✔
DDB2	(-)	(-)	✔	(-)

## Data Availability

No new data were created or analyzed in this study.

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
