# Peer review of "CUL4-Based Ubiquitin Ligases in Chromatin Regulation: An Evolutionary Perspective"

_cells, 2025, doi:10.3390/cells14020063_

Round 1

Reviewer 1 Report

Comments and Suggestions for Authors

This manuscript on CRL4 ligases is extensive but lacks of focus and seems to be still at a draft stage. In addition, it is not clear which message the authors would like to convey.  At this stage I can not thus recommend this manuscript for publications, unfortunately.

I would suggest the authors to rewrite the manuscript extensively, by focusing it only on a subset of organisms and/or a subset of molecular outputs (for example: chromatin regulation), while providing further details and context. For example, the discussion of CRL4 in plants have many factual mistakes, lacks important details and citations; I would thus recommend to omit it, and focusing on other organisms(s).

Also the structure of CRL4 and their regulation is not explained in detail , and a figure representing a model can be included . The tables are too general to be informative. The whole manuscript should be indeed checked for figure and text accuracy, and proper citations. For example, the phylogenetic tree depicted in figure 1 is too general: it should be explained how it was built and which data is based on (see reference 13);  furthermore, primary literature and not reviews should be cited wherever possible. 

Author Response

Response to Reviewer #1:

We sincerely appreciate your thorough review of our manuscript. Your comments have been invaluable in enhancing the clarity and quality of our work. Below, we provide detailed responses and outline the corresponding revisions, which are highlighted in the resubmitted files.

Comment:

"This manuscript on CRL4 ligases is extensive but lacks focus and seems to still be in a draft stage. In addition, it is not clear which message the authors would like to convey. At this stage, I cannot recommend this manuscript for publication, unfortunately."

Response:

We deeply regret the lack of focus and the draft-like quality of the initial submission. In response, we have revised the manuscript, incorporating the feedback from all three reviewers. We believe the improved version addresses these concerns and is now suitable for consideration.

Comment:

"I would suggest the authors to rewrite the manuscript extensively, focusing it only on a subset of organisms and/or a subset of molecular outputs (for example: chromatin regulation), while providing further details and context. For example, the discussion of CRL4 in plants has many factual mistakes, lacks important details and citations; I would thus recommend omitting it and focusing on other organism(s)."

Response:

We have removed the discussion of CRL4 in plants and revised related sections accordingly, ensuring a more focused and accurate presentation of the topic.

Comment:

"Also, the structure of CRL4 and its regulation is not explained in detail, and a figure representing a model can be included. The tables are too general to be informative. The whole manuscript should indeed be checked for figure and text accuracy, and proper citations. For example, the phylogenetic tree depicted in Figure 1 is too general: it should be explained how it was built and which data it is based on (see reference 13); furthermore, primary literature and not reviews should be cited wherever possible."

Response:

The revised manuscript now includes a model of the CRL4 structure and its regulation, presented in Figure 2. The phylogenetic tree (previously in Figure 1) has been removed to improve focus and avoid generalizations. We have also ensured the manuscript is thoroughly checked for accuracy in figures, text, and citations. We aim to direct readers to review articles for background knowledge while citing primary literature on the subject of CRL4.

We hope that these revisions and detailed responses adequately address the reviewers’ concerns and significantly enhance the clarity and scientific rigor of our manuscript.

Reviewer 2 Report

Comments and Suggestions for Authors

The article entitled “CLU4-based ubiquitin ligases in chromatin regulation: an evolutionary perspective” is a review of the literature describing the functions of the CUL4-containing Cullin-Ring ubiquitin ligases complexes in various organisms including eucaryote unicellular microorganism (trypanosoma), worms, yeasts, plants, flies and mammals. The authors concluded by a comparison and evolutionary analysis of the functions of CRL4.

In general, I found the article well documented, well organized and well written and I have only minor suggestions.

Chapter 2. CRL4 structure: A Figure illustrating the CRL4 structure could help in understanding line 107-112.

It appears that CRL4-mediated ubiquitination mainly leads to substrate degradation. It could be interesting to specify or discuss the type of ubiquitination carried out by CRLs. Can CRLs mediate non-degradative ubiquitination?

Author Response

Response to Reviewer #2:

We sincerely thank you for your constructive comments, which have greatly contributed to the refinement of our manuscript. Detailed responses to your suggestions are provided below, with corresponding revisions highlighted in the resubmitted files.

Comment:

"Chapter 2. CRL4 structure: A figure illustrating the CRL4 structure could help in understanding lines 107–112."

Response:

A model illustrating the CRL4 structure and regulation has been included as Figure 2 in the revised manuscript.

Comment:

"It appears that CRL4-mediated ubiquitination mainly leads to substrate degradation. It could be interesting to specify or discuss the type of ubiquitination carried out by CRLs. Can CRLs mediate non-degradative ubiquitination?"

Response:

CRL4A/BDDB2-mediated ubiquitylation of histones H3 and H4 facilitates chromatin loosening without degradation during DNA repair. This suggests that degradation is not the sole outcome of CRL4-mediated ubiquitylation. We have now included a discussion on non-degradative ubiquitination mediated by CRL4. “Notably, CRL4 substrates are not always directed towards degradation [130], suggesting that ubiquitylated neo-substrates targeted by CRL4-based TPD (targeted protein degradation) may exhibit unintended secondary functions. We believe it is crucial to investigate the regulatory mechanisms governing the ubiquitylation of CRL4 substrates in greater detail. For instance, it is important to determine whether the specific structure of the substrate, substrate-specific deubiquitylases, or CRL4 itself influences the type and length of the ubiquitin chain” (line 797-803).

We hope that these revisions and detailed responses adequately address the reviewers’ concerns and significantly enhance the clarity and scientific rigor of our manuscript.

Reviewer 3 Report

Comments and Suggestions for Authors

The review article by Nakagawa and Nakagawa summarizes the nuclear roles of CUL4 in a variety of organisms to demonstrate its evolutionary conservation of function. While the title indicates a focus on chromatin regulation, the abstract is more broad and speaks of both the nuclear and cytoplasmic roles of CUL4B and CUL4A, respectively. It also speaks of different nuclear and cytoplasmic substrates and poses that the review will “discuss the conserved and novel functions of CRL4s.” This leads the reader to assume that cytoplasmic roles and substrates will be discussed. Absence of this discussion is disappointing and makes the abstract feel misleading.

Without discussion of the cytoplasmic functions and substrates of CUL4 (and accordingly the differences between CUL4A and CUL4B), this review is fairly standard and covers little new ground. To increase impact and interest, I recommend a revision that includes these topics. Moonlighting functions of traditionally nuclear or cytoplasmic proteins are becoming increasingly prevalent, and this is currently a hot topic.

Author Response

Response to Reviewer #3:

We are deeply grateful for your thoughtful review and insightful suggestions. Your feedback has been instrumental in improving the depth and impact of our manuscript. Below, we provide detailed responses to your comments, along with the corresponding revisions highlighted in the resubmitted files.

Comment:

"The review article by Nakagawa and Nakagawa summarizes the nuclear roles of CUL4 in a variety of organisms to demonstrate its evolutionary conservation of function. While the title indicates a focus on chromatin regulation, the abstract is broader and discusses both nuclear and cytoplasmic roles of CUL4B and CUL4A, respectively. It also mentions different nuclear and cytoplasmic substrates and suggests that the review will 'discuss the conserved and novel functions of CRL4s.' This leads the reader to assume that cytoplasmic roles and substrates will be discussed. Absence of this discussion is disappointing and makes the abstract feel misleading.

Without discussion of the cytoplasmic functions and substrates of CUL4 (and accordingly the differences between CUL4A and CUL4B), this review is fairly standard and covers little new ground. To increase impact and interest, I recommend a revision that includes these topics. Moonlighting functions of traditionally nuclear or cytoplasmic proteins are becoming increasingly prevalent, and this is currently a hot topic."

Response:

We acknowledge the reviewer’s point and recognize the importance of discussing cytoplasmic functions of CRL4. However, to maintain focus on the nuclear roles of CRL4, which is the primary topic of this review, we did not include an extensive discussion on cytoplasmic functions. Nevertheless, to highlight their significance, we have added a sentence discussing the involvement of CRL4 in autophagy, as an example. “To achieve a comprehensive evolutionary perspective on CRL4, the elucidation of its non-chromatin and cytosolic functions is indispensable. For instance, numerous studies have demonstrated the involvement of CRL4 in autophagy through the ubiquitylation of key substrates, including Beclin 1 [205], DCAF3/AMBRA1 [206], WIPI2 [207], MAGE-A3/6 [208] and LAMP2 [209], thereby establishing a connection between the two protein degradation systems. (line 806-810).

We hope that these revisions and detailed responses adequately address the reviewers’ concerns and significantly enhance the clarity and scientific rigor of our manuscript.

Round 2

Reviewer 3 Report

Comments and Suggestions for Authors

The authors did not address my concerns adequately, one sentence does not make a discussion. I still feel lack of discussion of cytoplasmic function significantly decreases the novelty and interest of the review and could have easily been added with little extra work, but I will not object to its publication.